# Semiconductor Optical Amplifiers with Wide Gain Bandwidth and Enhanced Polarization Insensitivity Based on Tensile-Strained Quantum Wells

**DOI:** 10.3390/s24113285

**Published:** 2024-05-21

**Authors:** Hui Tang, Meng Zhang, Changjin Yang, Lei Liang, Li Qin, Yuxin Lei, Peng Jia, Yongyi Chen, Yubing Wang, Yue Song, Cheng Qiu, Yuntao Cao, Dabing Li, Lijun Wang

**Affiliations:** 1State Key Laboratory of Luminescence and Applications, Changchun Institute of Optics, Fine Mechanics and Physics, Chinese Academy of Sciences, Changchun 130033, China; tanghui21@mails.ucas.ac.cn (H.T.); zhangmeng223@mails.ucas.ac.cn (M.Z.); yangchangjin20@mails.ucas.ac.cn (C.Y.); qinl@ciomp.ac.cn (L.Q.); leiyuxin@ciomp.ac.cn (Y.L.); jiap@ciomp.ac.cn (P.J.); wangyubing@ciomp.ac.cn (Y.W.); songyue@ciomp.ac.cn (Y.S.); qiucheng@ciomp.ac.cn (C.Q.); lidb@ciomp.ac.cn (D.L.); wanglj@ciomp.ac.cn (L.W.); 2Daheng College, University of Chinese Academy of Sciences, Beijing 100049, China; 3Peng Cheng Laboratory, No. 2, Xingke 1st Street, Shenzhen 518000, China; 4Jilin Changguang Jixin Technology Co., Ltd., No. 206, Software Road, HTDZ, Changchun 130022, China; 5Jlight Semiconductor Technology Co., Ltd., No. 1588, Changde Road, ETDZ, Changchun 130102, China; 6National Key Laboratory of Advanced Vehicle Integration and Control, China FAW Corporation Limited, Changchun 130000, China

**Keywords:** semiconductor optical amplifier, wide gain bandwidth, low polarization, low noise, optical communication

## Abstract

The paper presents a wide-bandwidth, low-polarization semiconductor optical amplifier (SOA) based on strained quantum wells. By enhancing the material gain of quantum wells for TM modes, we have extended the gain bandwidth of the SOA while reducing its polarization sensitivity. Through a combination of tilted waveguide design and cavity surface optical thin film design, we have effectively reduced the cavity surface reflectance of the SOA, thus decreasing device transmission losses and noise figure. At a wavelength of 1550 nm and a drive current of 1.4 A, the output power can reach 188 mW, with a small signal gain of 36.4 dB and a 3 dB gain bandwidth of 128 nm. The linewidth broadening is only 1.032 times. The polarization-dependent gain of the SOA is below 1.4 dB, and the noise figure is below 5.5 dB. The device employs only I-line lithography technology, offering simple fabrication processes and low costs yet delivering outstanding and stable performance. The designed SOA achieves wide gain bandwidth, high gain, low polarization sensitivity, low linewidth broadening, and low noise, promising significant applications in the wide-bandwidth optical communication field across the S + C + L bands.

## 1. Introduction

With the explosive growth of network traffic in recent years, optical communication technology has imposed higher demands on bandwidth and speed. Currently, one of the main optical communication bands is the C-band, centered around 1550 nm. This band is characterized by low loss in optical fiber transmission and falls within the range of wavelengths with lower risk of eye damage. However, as the demand for data transmission capacity continues to increase, the channel capacity based on the C-band has become inadequate to meet the rapidly growing requirements. Therefore, further expanding the channel capacity of optical communication has become urgent [1,2,3].

Expanding the optical communication bands to include the L-band and S-band, which also fall within the optical fiber transparent window, is currently an effective solution. In 2020, the research group led by Jeremie Renaudier achieved a breakthrough by implementing a 100 nm multi-bandwidth wavelength-division multiplexing (WDM) in the C + L bands. This enabled the transmission of 250 communication channels with a total capacity of 115.9 Tb/s over optical fiber. The channel spacing was 50 GHz, and the transmission distance exceeded 100 km, making it the longest-distance optical fiber transmission system at the time, with a 100 nm channel width [4]. In 2024, the research team led by Tomoyuki Kato achieved the independent high-speed transmission of dense wavelength division multiplexing (DWDM) signals and 400GbE signals on a single optical fiber [5]. The team assembled 176-channel DWDM signals spanning the S + C + L bands, enabling a remarkable 400 Gb/s fiber optic transmission within a 200 nm bandwidth range. Broadening the channel bandwidth not only enhances the transmission capacity but also widens the intervals between channels, thereby alleviating the demands on signal source linewidth and quality in the communication process.

However, covering the bandwidth of the C + L bands or even the S + C + L bands poses significant challenges for optical amplifiers. Currently, erbium-doped fiber amplifiers (EDFAs) typically have narrow gain bandwidths, often struggling to reach 60 nm. Adjusting the gain bandwidth of EDFAs is difficult, and even if they can cover different bands, they often fail to meet the future demands of integrated, low-cost, and low-power optical amplifiers for optical communication [6,7]. In contrast, SOAs boast compact dimensions, low power consumption, and possess the flexibility to finely tune the gain bandwidth via energy band engineering Moreover, the current reported gain bandwidth has been able to reach 80 nm~100 nm [8,9], which makes it more suitable as an amplifier for broadly tuned light sources [10,11,12].

This paper proposes a 1550 nm high-power, wide-gain spectrum, low-polarization SOA based on strained quantum well structures. By enhancing the material gain for TM modes, we effectively reduce the polarization-dependent gain (PDG) of the SOA. Furthermore, the strained, flat material gain spectrum expands the gain bandwidth of the SOA. We simulated and designed suitable waveguide tilt angles and anti-reflective (AR) optical thin-film structures, effectively reducing the cavity surface reflectance of the SOA through their combined use. The second section of the paper discusses the structural design and fabrication process of the SOA. The third section conducts performance testing and analysis of the SOA, while the fourth section analyzes and discusses the performance indicators of the SOA. Finally, the fifth section provides a summary of the work presented in this paper.

## 2. SOA Structure Design and Preparation Process

(1)SOA Structure Design

To achieve a wide gain bandwidth and low polarization in the SOA, we devised a narrow-ridge tilted waveguide SOA, as shown in Figure 1a. The device measures 2.5 mm in length and 0.5 mm in width. The epitaxial structure was grown using a metal–organic chemical vapor deposition (MOCVD) system on an InP substrate to ensure high-quality epitaxy, as shown in Figure 1b.

To reduce the reflectance of the SOA and thus decrease transmission losses and noise, we employed a tilted waveguide structure design. The naturally cleaved facet reflectance of a straight vertical output waveguide is approximately 32%. However, in a tilted waveguide, the active region deviates from the cleavage plane center, which can theoretically reduce the facet reflectance by two orders of magnitude. In the ideal scenario of a smooth cross-section, the facet reflectance R_facet_ of a tilted waveguide SOA can be approximately expressed as follows [13]:(1)$$R_{facet}=R_{f}\exp[-2{\left(\pi n_{2}w\theta /\lambda \right)}^{2}],$$
(2)$$R_{f}={\left(\frac{n_{1}\cos\theta -\sqrt{1-{n_{1}}^{2}\sin^{2}\theta }}{n_{1}\cos\theta +\sqrt{1-{n_{1}}^{2}\sin^{2}\theta }}\right)}^{2},$$
where θ is the waveguide tilt angle, n_1_ = 3.4018 is the effective refractive index of the active region, n_2_ = 3.164 is the effective refractive index of the cladding, w is the waveguide width, and λ is the wavelength of the signal light. As shown in Figure 2, the tilt angle of the waveguide is inversely proportional to the width of the waveguide. As the waveguide width increases, the tilt angle required to reduce the facet reflectance decreases.

However, an excessively wide waveguide can also affect the single-mode performance of the SOA. To maintain single-mode operation, the etch depth of the waveguide is inversely related to the waveguide width. Deeper etch depths make it easier to encounter higher-order modes under the same ridge width conditions. Additionally, the etch depth also influences the optical field confinement factor of the active region. Shallow etch grooves can severely impact the power of the SOA. Taking into account the thicknesses of the active region, upper waveguide layer, and upper cladding layer, an appropriate etch depth for the epitaxial structure designed in this paper is around 1600 μm. This depth allows for a single-mode operation with a large ridge width of 6 μm, ensuring a single-mode operation within acceptable error margins.

Taking all these factors into consideration, we opted for the commonly used tilt angle of 7° for the waveguide, as reported in the literature. At this angle, under ideal conditions, the facet reflectance of waveguides with widths of 4 μm, 5 μm, and 6 μm can all be reduced to below 0.01%. In subsequent comparative experiments, we observed differences in the maximum output power among devices with waveguide widths of 4 μm, 5 μm, and 6 μm. Specifically, the device with a waveguide width of 4 μm had a maximum output power of approximately 70.6 mW, the device with a waveguide width of 5 μm had a maximum output power of approximately 139.3 mW, and the device with a waveguide width of 6 μm had a maximum output power of 188 mW. Consequently, we ultimately chose a waveguide structure with a width of 6 μm and a tilt angle of 7°.

(2)Epitaxial structure design

To achieve low polarization and wide-spectrum SOA, we incorporated strained quantum wells with a matching potential barrier structure in the active region. This approach effectively increases the energy of the light hole sub-band while reducing the difference with the energy of the heavy hole sub-band, thus balancing the TE and TM mode gains of the SOA. At the center of the Brillouin zone in the active region, at k = 0, the band edge energies of the heavy hole and light hole bands can be expressed as follows:(3)$$E_{HH}(k=0)=a_{v}(\varepsilon_{x}+\varepsilon_{y}+\varepsilon_{z})-\frac{b}{2}(\varepsilon_{x}+\varepsilon_{y}-2\varepsilon_{z}),$$
(4)$$E_{HH}(k=0)=a_{v}(\varepsilon_{x}+\varepsilon_{y}+\varepsilon_{z})+\frac{b}{2}(\varepsilon_{x}+\varepsilon_{y}-2\varepsilon_{z}),$$

The band edge energy of the electron conduction band can be expressed as follows:(5)$$E_{c}(k=0)=E_{g}+a_{c}(\varepsilon_{x}+\varepsilon_{y}+\varepsilon_{z}),$$
where a_v_ and a_c_ are the conduction band deformation potential and valence band deformation potential, respectively, b is the tangential strain deformation potential, and ε_x_, ε_y_, and ε_z_ are the lattice mismatch quantities. Furthermore, the net transition energy from the conduction band to the heavy hole and light hole bands can be expressed, respectively, as follows:(6)$$E_{c-HH}=E_{g}+(a_{c}-a_{v})(\varepsilon_{x}+\varepsilon_{y}+\varepsilon_{z})-\frac{b}{2}(\varepsilon_{x}+\varepsilon_{y}-2\varepsilon_{z}),$$
(7)$$E_{c-LH}=E_{g}+(a_{c}-a_{v})(\varepsilon_{x}+\varepsilon_{y}+\varepsilon_{z})+\frac{b}{2}(\varepsilon_{x}+\varepsilon_{y}-2\varepsilon_{z}),$$
where E_g_ is the band gap in the strain-free case.

The above discussion elucidates the impact of strain on the energy bands of quantum wells. Next, we can derive the relationship between strained quantum wells and gains for different polarization states. In the ideal scenario where carrier losses are not considered, the material gain of the SOA can be expressed as follows [14]:(8)$$g(\hbar \nu )=\frac{{m_{r}}^{*}e^{2}}{n_{r}c\varepsilon_{0}{m_{0}}^{2}\nu \hbar^{2}L_{z}}{\int_{0}^{\infty }dE_{t}|\overset{\wedge }{e}*p_{cv}|}^{2}\frac{\gamma /\pi }{{\left[E_{h}^{e}+E_{t}-\hbar \nu \right]}^{2}+\gamma^{2}}\times \left[f_{c}\left(E_{t}\right)-f_{v}\left(E_{t}\right)\right],$$
(9)$$\frac{1}{{m_{r}}^{*}}=\frac{1}{{m_{e}}^{*}}+\frac{1}{{m_{h}}^{*}},$$
(10)$$f_{c}(E_{t})=1/(1+\exp\left[\left(E_{g}+E_{e}+\frac{{m_{r}}^{*}}{{m_{e}}^{*}}E_{t}-F_{c}\right)/\left(k_{B}T\right)\right]),$$
(11)$$f_{v}(E_{t})=1/(1+\exp\left[\left(E_{h}-\frac{{m_{r}}^{*}}{{m_{h}}^{*}}E_{t}-F_{v}\right)/\left(k_{B}T\right)\right]),$$
where ℏ is the reduced Planck constant, ν is the input signal optical frequency, e is the electron energy, n_r_ is the refractive index, c is the speed of light in a vacuum, ε_0_ is the vacuum permittivity, m_0_ is the electron mass, L_z_ is the thickness of the quantum well, ${\left|\hat{e}*p_{cv}\right|}^{2}$ is the matrix elements of the momentum operator, γ is the refractive index diffusion factor, E_t_ is the energy level composite center, $E_{h}^{e}$ is the band-side jump energy, $m_{e}^{*}$ and $m_{h}^{*}$ are the approximate masses of the conduction band electrons and valence band holes, k_B_ is Boltzmann’s constant, T is the reference temperature, and E_e_ and E_h_ are the energy levels of the electrons and holes after the strains found above.

Matrix elements of the momentum operator ${\left|\hat{e}*p_{cv}\right|}^{2}$ is polarization dependent and directly affects the gain of the quantum well for different polarization states. For TE polarization, considering that the light field is polarized along the x or y direction, so that $\hat{e}=\hat{x}$, we take the average of the matrix elements of the optical dipole moment with respect to the azimuthal angle ϕ of the quantum well plane. The matrix element for the transition from the electron with spin up <iS↑′| to the heavy hole band can be expressed as follows:(12)$$|\hat{e}\cdot p_{cv}{|}^{2}=\langle |\hat{e}\cdot M_{c-hh}{|}^{2}\rangle =\frac{1}{2\pi }\int_{0}^{2\pi }d\phi {\left|\hat{x}\cdot M_{c-hh}\right|}^{2}=\frac{3}{4}(1+\cos^{2}\theta ){M_{b}}^{2},$$
(13)$$\cos^{2}\theta_{nm}=\frac{E_{en}+\left|E_{hm}\right|}{E_{en}+\left|E_{hm}\right|+\frac{\hbar^{2}{k_{t}}^{2}}{2{m_{r}}^{*}}},$$
where m_0_ is the electron mass and M_b_ is the matrix elements of momentum operator. Similarly, the matrix element for the transition from the electron with another spin down <iS↓′| in the conduction band to the light hole band can be expressed as follows:(14)$$\langle |\hat{e}\cdot M_{c-lh}{|}^{2}\rangle =\frac{1}{2\pi }\int_{0}^{2\pi }d\phi \left({\left|{\langle iS{↓}^{\prime }\left|p_{x}\right|\frac{3}{2},\frac{1}{2}\rangle }^{\prime }\right|}^{2}+{\left|{\langle iS{↓}^{\prime }\left|p_{x}\right|\frac{3}{2},-\frac{1}{2}\rangle }^{\prime }\right|}^{2}\right)=(\frac{5}{4}-\frac{3}{4}\cos^{2}\theta ){M_{b}}^{2},$$

For TM polarization, let $\hat{e}=\hat{z}$, and the matrix element for the transition from the electron with spin up <iS↑′| to the heavy hole band can be expressed as follows:(15)$$\langle |\hat{e}\cdot M_{c-hh}{|}^{2}\rangle =\frac{1}{2\pi }\int_{0}^{2\pi }d\phi {\left|\hat{z}\cdot M_{c-hh}\right|}^{2}=\frac{3}{2}\sin^{2}\theta {M_{b}}^{2},$$

The matrix element for the transition from another spin-down electron <iS↓′| in the conduction band to the light hole band can be expressed as follows:(16)$$\langle |\hat{e}\cdot M_{c-lh}{|}^{2}\rangle =\frac{1}{2\pi }\int_{0}^{2\pi }d\phi \left({\left|{\langle iS{↓}^{\prime }\left|p_{z}\right|\frac{3}{2},\frac{1}{2}\rangle }^{\prime }\right|}^{2}+{\left|{\langle iS{↓}^{\prime }\left|p_{z}\right|\frac{3}{2},-\frac{1}{2}\rangle }^{\prime }\right|}^{2}\right)=\frac{1+3\cos^{2}\theta }{2}{M_{b}}^{2},$$

Substituting Equations (12) and (14)–(16) back into Equation (8), the material gain g_pol_ of the quantum well for different polarization states can be obtained. The material gain curves for the compressive strain quantum well and tensile strain quantum well are shown in Figure 3. Comparatively, the TM mode gain curve, TM_ts, of strained quantum wells exhibits a significant improvement over the TM mode gain curve, TM_ps, of unstrained quantum wells. The gain at the center position of the gain curve is now close to that of the TE mode gain in strained quantum wells, indicating similar material gains for both TE and TM modes in strained quantum wells. Consequently, the polarization sensitivity of the SOA has been significantly enhanced in strained quantum wells. However, conversely, the TE mode gain curve, TE_ts, of strained quantum wells is lower than the TE mode gain curve, TE_ps, of unstrained quantum wells, suggesting that the overall gain of the strained quantum well structure is lower than that of the unstrained quantum well structure. Additionally, the enhancement of strained quantum well TM gain leads to a flatter gain curve when the TE mode gain curve and TM mode gain curve are overlapped. This results in a wider gain bandwidth for the SOA compared to unstrained quantum wells.

Table 1 presents the detailed epitaxial structure. The active region consists of 5 AlGaInAs quantum wells and 6 AlGaInAs barriers with a strain level of −0.21%. Positioned between the n-type AlGaInAs lower waveguide layer with a three-layer compositional gradient and the AlGaInAs upper waveguide layer with a two-layer compositional gradient, this active region structure has a distinctive feature: a lower optical field confinement factor, forming a weak waveguide structure. This feature facilitates two key actions: firstly, it broadens the optical field mode towards both the n-type lower waveguide layer and the p-type upper waveguide layer. Secondly, it deviates the central optical field mode away from the active region. Consequently, this configuration enhances the saturation output power.

(3)Cavity surface optical film design

Although an idealized slanted waveguide structure can achieve extremely low facet reflectance, in practical implementation, surface roughness can affect facet reflectance. Therefore, to achieve extremely low reflectance, it is also necessary to deposit AR optical films on the facet. Similar to the purpose of the tilted waveguide structure design, the AR optical film on the SOA facet aims to reduce the facet reflectance, thus lowering SOA losses and noise while optimizing its performance. Due to the high refractive index of semiconductor materials, typically ranging from 3 to 4, the reflectance of the facet is usually around 32%. Therefore, to reduce the facet reflectance, an AR optical film design is applied to the SOA facet. When the signal light passes through the AR film and enters the active region of the SOA from the air, the characteristic matrix of the AR film can be represented as follows [15]:(17)$$E_{0}\left[\begin{array}{c} 1\\ Y \end{array}\right]=\left[\begin{array}{cc} \cos\delta_{1} & \frac{i}{\eta_{1}}\sin\delta_{1}\\ i\eta_{1}\sin\delta_{1} & \cos\delta_{1} \end{array}\right]\left[\begin{array}{c} 1\\ \eta_{2} \end{array}\right]E_{2},$$
where $\delta_{1}=\frac{2\pi }{\lambda }n_{b}d_{1}\cos\theta_{1}$ is the phase thickness of the optical film, θ_1_ is the angle between the incident light and the normal to the incident plane, d_1_ is the optical thickness of the film, n_b_ is the effective refractive index of the optical film. η_1_ and η_2_ are the optical admittance of the optical film and the active layer of the SOA, respectively, and E_0_ and E_2_ are the light field in air and the light field in the SOA, respectively. In the optical thin film, η_1_ is polarization-dependent. For TE mode, η_1_ can be expressed as follows:(18)$$\eta_{1\_TE}=n_{b}\cos\theta_{1}.$$

For TM modes, η_1_ can be expressed as follows:(19)$$\eta_{1\_TM}=n_{b}/\cos\theta_{1}.$$

Further we can obtain the cavity surface energy reflectivity R of the SOA as follows:(20)$$R=\frac{\eta_{0}-Y}{\eta_{0}+Y}\cdot {\left(\frac{\eta_{0}-Y}{\eta_{0}+Y}\right)}^{*}=\frac{{\left(\eta_{0}-\eta_{2}\right)}^{2}\cos^{2}\delta_{1}+{\left(\eta_{0}\eta_{2}/\eta_{1}-\eta_{1}\right)}^{2}\sin^{2}\delta_{1}}{{\left(\eta_{0}+\eta_{2}\right)}^{2}\cos^{2}\delta_{1}+{\left(\eta_{0}\eta_{2}/\eta_{1}+\eta_{1}\right)}^{2}\sin^{2}\delta_{1}},$$

From Equation (21), it can be seen that a range of extreme values of R exists at optical film thicknesses ndcosθ of integer multiples of 1/4 wavelength. Substituting Equations (19) and (20) into (21), we can obtain the simulation curves of reflectivity for different polarization states in a single-layer AR film, as shown in Figure 4. From the reflection curves of TE and TM modes, it is evident that there is a certain disparity in the reflectance compared to the average reflectance curve at a 7° tilt angle. Specifically, the low reflectance band of the TE mode is narrower than that of the TM mode, indicating that losses in the TE mode will be greater than those in the TM mode at the boundaries of the gain bandwidth. While the average reflectance can achieve nearly 0.1% reflectance at the central wavelength, the wavelength range with reflectance below 0.1% is only around 45 nm, which cannot meet the amplification requirements of SOAs with wide bandwidths. Therefore, to meet the demands of SOAs with wide bandwidths, it is necessary to employ multi-layer ultra-broadband AR films.

Multi-layer AR coatings are typically composed of alternating layers of materials with different refractive indices to overcome refractive index limitations. From Equation (19), the characteristic matrix for a K-layer AR coating can be further expressed as follows:(21)$$E_{0}\left[\begin{array}{c} 1\\ Y \end{array}\right]=\left\{\prod_{j=1}^{K}\left[\begin{array}{cc} \cos\delta_{j} & \frac{i}{\eta_{j}}\sin\delta_{j}\\ i\eta_{j}\sin\delta_{j} & \cos\delta_{j} \end{array}\right]\right\}\left[\begin{array}{c} 1\\ \eta_{s} \end{array}\right]E_{K+1},$$

The reflection model for multilayer AR coatings is highly intricate, and its reverse design necessitates continuous optimization of numerical parameters to generate membrane structures that meet design requirements. The material system for the ultra-broadband AR film designed in this study employs Ti_3_O_5_ and SiO_2_, with the effective refractive indices of materials illustrated in Figure 5a. The AR optical film structure was simulated and designed using tfcalc (ver 3.5) software, and its reflection simulation curve is depicted in Figure 5b. It achieves energy reflection rates below 0.1% over a wavelength range from 1468 nm to 1600 nm, totaling 132 nm, effectively covering the anticipated gain bandwidth for our SOA.

To verify whether the designed AR film meets expectations in terms of transmittance, we employed the on-chip modulation index method to test the transmittance of our designed AR film. The on-chip modulation index method is a technique used to estimate the reflectance of the AR optical film on its cavity surface using the gain model of a superluminescent diode (SLD). The modulation index m of the SLD is expressed as follows:(22)$$m=\frac{P_{\max}-P_{\min}}{P_{\max}+P_{\min}},$$
where P_max_ and P_min_ represent the maximum and minimum values of the SLD output power density spectrum. Additionally, according to the gain model of SLD, m can be expressed as follows:(23)$$m=\frac{2\left|a\right|}{1+{\left|a\right|}^{2}},$$
where a represents the round-trip gain factor, which can be expressed as follows:(24)$$a=\sqrt{R_{1}R_{2}}e^{2j\varphi L}e^{G_{l}},$$
where R_1_ and R_2_ are the cavity surface reflectivity, L is the length of the laser, and G_l_ is the single-pass gain, which can be expressed as follows:(25)$$G_{l}=(\kappa \frac{I}{I_{t}}-\alpha )L,$$
where α is the loss coefficient, I_t_ is the threshold current of the laser, and κ is the gain coefficient at I_t_, From Equations (24) and (25), the round-trip gain factor obtained a can be expressed as follows:(26)$$\ln\left|a\right|=\left(\kappa L\right)\frac{I}{I_{t}}-\left(\kappa L-\ln\frac{\sqrt{R_{1}R_{2}}}{R_{s}}\right),$$
where R_s_ is the cavity surface reflectivity in the case of uncoated surface, which is usually 0.32. From Equations (22), (23) and (26), we can take I/I_t_ at different threshold currents as the independent variable and ln|a| as the dependent variable and take a number of points with I/It < 1.5 for straight line fitting. The cavity surface reflectivity of the AR film can be obtained from the slope k and the intercept b of the straight line, which can be expressed as follows:(27)$$R_{1}R_{2}={R_{s}}^{2}\exp\left(2(k+b)\right)$$

Hence, we isolated individual tubes of varying lengths and conducted tests on their current and power characteristics subsequent to coating one side with an AR film and the other side with a high-reflectance (HR) film boasting 98% reflectivity. From these tubes, a singular tube with a threshold current (It) of 60 mA was selected for further analysis. The output optical power spectrum was then measured at current values of 85 mA, 90 mA, 95 mA, 100 mA, 105 mA, and 110 mA. The results of straight-line fitting to the data are shown in Figure 6. The outcome of the linear regression analysis reveals the coefficients k = 3.896 and b = −7.33. By incorporating Equation (27), we can deduce the reflectance of the AR film, denoted as R1, to be 0.0108%. This value harmonizes with the anticipated specifications outlined in the design phase.

(4)SOA production

The actual structural diagram of the SOA under test is shown in Figure 7a, while its cross-sectional morphology is shown in Figure 7b, as observed under a scanning electron microscope. The tabletop of the waveguide exhibits a flat etched surface, with a side slope measuring 89.52°. The epitaxial wafer of the SOA is grown in a single step using MOCVD, and the waveguide structure is fabricated using an I-line photolithography machine. This design avoids the complexity and cost escalation associated with secondary epitaxy, while achieving excellent SOA patterns and stable performance solely through I-line photolithography. The p-type electrode and n-type electrode of the SOA employ Ti/Pt/Au and Au/Ge/Ni/Au materials, respectively. As illustrated in Figure 7c, the SOA is flip-chip bonded with its n-side facing downwards onto an aluminum nitride heat sink, establishing ohmic contact.

## 3. SOA Performance Testing and Analysis

(1)Test platform

During the testing process, the heat sink was fixed onto a thermoelectric cooler (TEC) to ensure precise temperature control during testing. The testing platform for the SOA under examination is shown in Figure 8. Light generated by the tunable laser passes through a polarization controller and is then coupled into the waveguide via a tapered optical fiber. At the other end, the light was coupled out via a lens and directed to various measurement devices, including optical power meters, spectrometers, and linewidth measurement systems, to evaluate parameters such as output power, gain, amplified spectrum, and linewidth broadening of the tested SOA.

(2)Amplified spontaneous emission spectroscopy test

In the experiment, we investigated the performance of the SOA device by controlling the temperature and driving current. Initially, under TEC temperature conditions of 20 °C, we measured the amplified spontaneous emission (ASE) spectrum of the SOA under test. As shown in Figure 9a, the ASE spectrum was centered around 1520 nm. With an increase in the driving current, the ASE spectrum gradually broadened, and there was a noticeable increase in ASE power in the longer wavelength region. Subsequently, we set the driving current to 1 A and varied the TEC temperature to 10 °C, 20 °C, 30 °C, and 40 °C, respectively, to test the change in ASE power of the SOA. As shown in Figure 9b, the ASE power of the SOA significantly decreased with increasing temperature. With the increase in current or temperature, the ASE spectrum underwent a redshift. As previously discussed during the quantum well design, the material gain center was biased towards shorter wavelengths for this reason. Following the redshift, the ASE spectrum center approached 1550 nm, consistent with expectations.

Additionally, the ripple in the ASE spectrum of the tested SOA was small, especially when the driving current is 1.4 A, with only 0.042 dB. This indicated that the tilted waveguide design effectively reduces the cavity surface reflectivity of the SOA. Moreover, the specially designed AR optical thin-film structure also effectively reduces the cavity surface reflectivity of the SOA, achieving reflectivity below 0.1%. This extremely low cavity surface reflectivity not only reduced the ripple in spontaneous emission but also effectively suppressed the spontaneous noise of the SOA. Therefore, it can be observed that the adoption of tilted waveguide design and cavity surface coating can improve the spectral characteristics of the SOA.

(3)Single-wavelength output power and gain test

In the experiment, we configured the laser wavelength to 1550 nm and regulated the input signal power to 6 dBm. Subsequently, we tuned the TEC temperature to 15 °C, 20 °C, 25 °C, and 30 °C, respectively. We then proceeded to examine the drive current and output power profiles of the amplifier, as depicted in Figure 10a. The experimental findings unveil that the SOA enters the domain of linear amplification as the driving current escalates to 0.1 A, achieving a stabilized gain. The power exhibits linear growth, gradually approaching saturation after the driving current surpasses 1 A. The SOA demonstrates heightened stability at 15 °C, 20 °C, and 25 °C, yet evinces a conspicuous decline in power at 30 °C. Specifically, the output power of the SOA registers at 188 mW when the driving current reaches 1.4 A under a TEC temperature of 15 °C.

By configuring the input current to 1 A and employing signal light wavelengths of 1530 nm, 1540 nm, 1550 nm, 1560 nm, and 1570 nm, we were able to generate a relationship curve between the output power and the gain, showcased in Figure 10b. Notably, when the output power reaches 16 dBm, the input power supplied to the SOA amounts to −20.4 dBm, resulting in an attainable maximum gain of 36.4 dB. As the input power rises to approximately 0 dBm, the output power reaches 22 dBm, with the gain decreasing to 21 dBm. When the output power reaches 22.63 dBm, the gain of the SOA drops to around 12 dB, indicating saturation.

The low optical confinement factor quantum well structure we employed allows the optical field within the waveguide to expand to the upper and lower waveguide layers, successively accommodating more optical field energy. This design partially compensates for the low output power caused by the strained quantum well structure, resulting in a higher output power of the SOA compared to previously reported low-polarization SOAs. However, despite these improvements, the output power remains insufficient compared to strain-compensated quantum well SOAs with similar waveguide structures.

(4)Spectrum and gain curve test after amplification

In the experiment, the driving currents were set to 0.5 A and 1 A, respectively, while the TEC temperature was maintained at 20 °C. The output power of the seed source laser was set to 6 dBm. By varying the wavelength of the seed source laser from 1480 nm to 1600 nm in increments of 5 nm, the amplified spectra of the SOA were recorded. As shown in Figure 11, the overlaid schematic represents the amplified spectra across the entire test band. The experimental results demonstrate excellent gain performance of the SOA within the 1480–1580 nm wavelength range. The side mode suppression ratio (SMSR) of the signal source used for testing approached 60 dB. After amplification by the amplifier, the SMSR of the SOA remained above 50 dB throughout the entire tuning range. The spectra amplified by the SOA exhibited low background noise across the entire wavelength range of 1480–1580 nm, with smooth curves attributed to the combination of the tilted waveguide structure and the ultra-wideband AR optical thin film. This integration resulted in ultra-low reflectivity across the entire gain bandwidth, effectively suppressing cavity reflections and reducing spontaneous noise.

In the experiment, the TEC temperature was controlled at 20 °C, and the wavelength of the seed source laser was set to powers of 0 dBm and 6 dBm, respectively, with a driving current of 1 A. Subsequently, by adjusting the wavelength of the seed source laser, the gain bandwidth of the amplifier was tested as shown in Figure 12. The experimental results indicate that at an input power of 0 dBm, the SOA under test was in a non-saturated state, with the gain across the entire bandwidth remaining unsaturated. When the driving current was increased to 1.25 A, the maximum gain of the SOA under test reached 21.768 dB.

When the input power reaches 6 dBm, the gain in the 1530 nm wavelength band, located at the center of the gain spectrum, approaches saturation, while the gain in the adjacent bands remains unsaturated. Consequently, the gain curve became smoother. With a driving current of 1.25 A, the maximum gain of the SOA under test reached 16.507 dB, with a 3 dB gain bandwidth of 128 nm. The device achieved a flat and wide gain spectrum, consistent with our expectations of extending the gain bandwidth by compensating for the device’s gain in the short wavelength band when designing the strained quantum well in the previous section. The ultra-wideband AR optical film we designed completely covers the entire gain bandwidth, ensuring that the device’s power is not lost due to cavity reflections within the gain bandwidth. The 3 dB gain bandwidth of the SOA covered the S-band, C-band, and parts of the L-band. As the current increases, the gain spectrum center of the SOA shifted towards longer wavelengths, with an increase in gain in the 1590 nm wavelength band. This indicated that the SOA could meet the amplification requirements of the S + C bands, but there was still room for improvement in the coverage of the gain spectrum in the L-band.

(5)Line width broadening characteristics test

As shown in Figure 13, we employed a delayed self-heterodyne linewidth measurement system to test the linewidth broadening of the SOA under test. After fitting the beat frequency peaks with Lorentzian profiles, half of the 3 dB bandwidth of the Lorentzian curve represents the linewidth of the SOA under test.

In the experiment, we set the power of the seed source laser to 6 dBm, drove the current to 1 A, and maintained the TEC temperature at 20 °C. Using a delayed self-heterodyne linewidth measurement system, we obtained and compared the linewidths of the laser source and the SOA output, as shown in Figure 14. According to the Lorentzian fitting results, which is the red line in the picture, the linewidth of the laser signal was 75.45 kHz, and after amplification by the amplifier, the linewidth of the output light was 77.9 kHz, indicating a mere 1.032-fold broadening of the signal light’s linewidth. A smaller linewidth broadening after amplification suggests that narrower channel spacing can be allowed for DWDM applications, allowing for more channels within the band. The SOA designed in this study exhibits minimal linewidth broadening, implying minimal impact on the linewidth of the signal light when used as an optical amplifier in communication systems, indicating lower requirements for the linewidth of the signal light source compared to conventional amplifiers.

(6)Polarization dependent gain test

In the experiment, we set the driving current to 1 A and the power of the seed source laser to 6 dBm. By adjusting the three-ring polarization controller, we controlled the polarization state of the input signal from the seed source laser. Through polarization-integrated sphere observation, we ensured that the input signal was in a linear polarization state. We selected the polarization states corresponding to maximum and minimum gains, which are necessarily two linear polarization states orthogonal to each other. Subsequently, we adjusted the driving current from 0.5 A to 1.4 A and observed the PDG as a function of the driving current, as shown in Figure 15a. Then, with the driving current set to 1 A, we varied the input signal wavelength from 1520 nm to 1580 nm and observed the PDG as a function of the input signal wavelength, as shown in Figure 15b.

The experimental results indicated that with an input signal of 6 dBm, the SOA operates in a region close to saturation. At this point, both TE and TM mode gains are near saturation, resulting in a PDG of 1.64 dB. As the current was further increased, the TE mode saturates more rapidly than the TM mode. Consequently, the difference between the TE and TM mode gains decreased further, leading to a reduction in PDG to less than 1.4 dB, with a minimum of 1.12 dB. By comparing the curves of TE mode, TM mode, and wavelength, it can be observed that at the gain center of 1530 nm, the gains of the TE and TM modes are comparable, with a PDG of less than 1.25 dB and a minimum of 0.89 dB. The design of the strained quantum well effectively balanced the gains of the TE and TM modes, thereby reducing the PDG of the SOA.

In summary, when the input signal is 6 dBm, the SOA operated in a state close to saturation and exhibited low polarization sensitivity. Its polarization-dependent gain was overall less than 1.4 dB, particularly when the SOA operated at the gain center of 1530 nm and was in a state of deep saturation at high currents. The polarization-dependent gain further decreased to below 1 db. This advantage of low polarization sensitivity in deep saturation states was not only suitable for use in fiber laser communications as an amplifier for the balanced amplification of signals with different polarization states but also for fiber gyroscopes. By employing SOA operating in the deep saturation region to suppress relative intensity noise, it enhanced the accuracy of fiber gyroscopes.

(7)Noise figure test

In the experiment, with the signal input power set at 6 dBm, the noise figure (NF) obtained through spectral analyzer testing is shown in Figure 16. The experimental results indicated that the NF of the SOA increases with temperature, attributed to the rise in spontaneous emission noise due to elevated temperature. At low driving currents, background noise remained inadequately suppressed, with NF primarily reflecting the SOA’s spontaneous emission noise. At a driving current of 0.1 A, NF exceeded 6.5 dB.

As the driving current increases, the SOA enters the linear amplification region, where the signal-spontaneous emission beat noise gradually becomes dominant, leading to a gradual suppression of background noise in the spectrum and, consequently, a noticeable reduction in SOA’s NF within the linear amplification region. At a temperature of 15 °C and a driving current of 0.5 A, the SOA exhibits a minimum NF of 4.904 dB. With a further increase in current, the SOA gradually enters the saturation region, where the high-power operating state amplifies the thermal current noise of the SOA. Hence, after the driving current exceeds 0.1 A, the previously steep decline in NF tends to increase to a certain extent with the increase in SOA driving current. Throughout the linear amplification and gain saturation regions, the SOA’s NF remained below 5.5 dB, with a minimum value of 4.904 dB, demonstrating the advantage of low NF.

## 4. Discussion

The SOA designed in this paper is compared with the SOA reported in the literature as shown in Table 2 [16,17,18,19,20]. Upon comparison, it is evident that the SOA designed in this study holds advantages in both output power and gain bandwidth. In the designed SOA, an output power of 188 mW can be achieved when the driving current is set to 1.4 A. With a laser power of 6 dBm, the SOA’s 3 dB gain bandwidth can reach 128 nm. Additionally, the SOA exhibits favorable polarization and noise characteristics. In near-saturation conditions, the polarization-dependent gain of the tested SOA is less than 1.4 dB, with a minimum polarization-dependent gain of 0.89 dB achievable. Furthermore, upon entering the linear amplification region, the NF of the tested SOA can be maintained below 5.5 dB, with a minimum NF of 4.904 dB.

It is evident that the structure of the strained quantum wells designed in our study effectively extends the gain bandwidth of the SOA and reduces the PDG of the SOA. Through the use of tilted waveguides and ultra-broadband AR optical films, we have ensured that the SOA maintains a cavity surface reflectivity of less than 0.1% across the entire gain bandwidth. This results in excellent performance of the SOA in terms of ASE ripple and NF. The 128 nm gain bandwidth of the SOA can meet the requirements of the S + C bands and part of the L band, offering broad prospects for application in wideband optical communication systems. Its advantages of low polarization sensitivity and low NF also extend its application areas to laser radar and fiber gyroscopes, distinguishing it significantly from currently available SOAs in the market.

However, there is still significant room for improvement in the maximum power of the SOA. While the strained quantum well structure has effectively enhanced the polarization-dependent gain, its impact on the maximum power and gain exceeds expectations. Similar SOA structures employing compressively strained quantum wells have achieved output powers exceeding 300 mW. This indicates that there is still ample room for improvement in the epitaxial structure of the SOA, necessitating a balance between TE and TM modal gains and enhancing the output power of the SOA. Additionally, the 3 dB gain bandwidth of the SOA has not fully covered the entire L band yet. To meet the bandwidth requirements of C + L band optical communication, further optimization of the quantum well structure is needed to shift the gain spectrum of the quantum wells towards longer wavelengths or to continue expanding the quantum well gain spectrum to achieve complete coverage of the C + L band. Complete coverage of the gain bandwidth across the C + L band will make the SOA more competitive in the field of optical communication.

## 5. Conclusions

In this paper, we have designed and fabricated a 1550 nm wide-gain-bandwidth, low-polarization-sensitivity, and low-noise tilted waveguide SOA based on strained quantum wells. Through the combined design of tilted waveguides and AR optical thin films, we have achieved a cavity surface reflectance of less than 0.1% across the entire gain bandwidth of the SOA, resulting in excellent device performance. The SOA designed in this paper can achieve an output power of 188 mW at a driving current of 1.4 A. When the seed source laser power is −20 dBm, the gain can reach 36.4 dB. The 3 dB gain bandwidth of the SOA can reach 128 nm, with a linewidth broadening factor of only 1.032. The polarization-dependent gain of the SOA can be as low as 0.89 dB, with a noise figure lower than 5.5 dB. The 128 nm gain bandwidth of this SOA can preliminarily meet the bandwidth requirements of S + C + L band optical communication systems, and it is expected to find wide application in the fields of optical communication and fiber optic gyroscopes. There is still room for improvement in the output power and gain bandwidth of the SOA, providing valuable directions and potential for future research efforts.

## Figures and Tables

**Figure 1 sensors-24-03285-f001:**
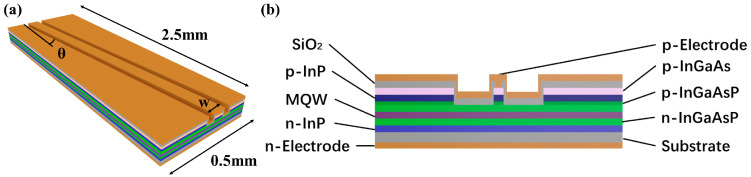
(**a**) Schematic of the structure of SOA; (**b**) schematic of the cross-sectional structure of SOA.

**Figure 2 sensors-24-03285-f002:**
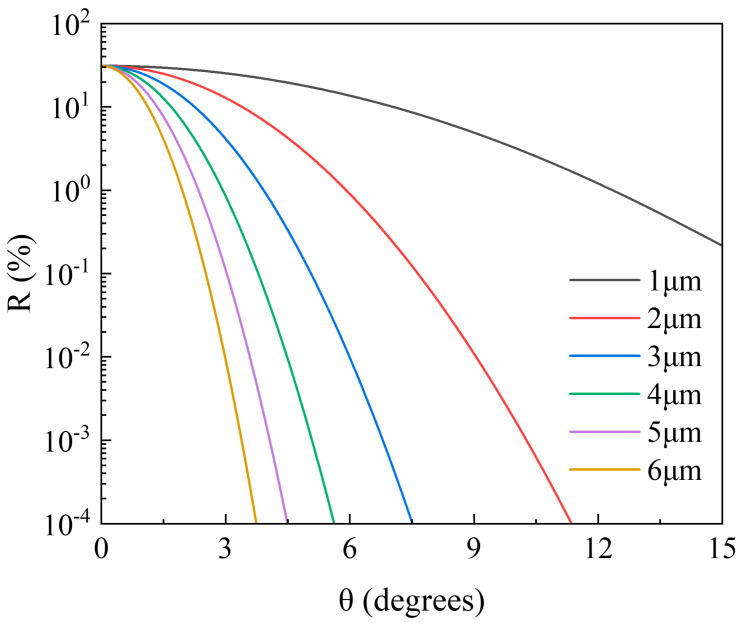
Waveguide tilt angle and cavity reflectance relationship curves.

**Figure 3 sensors-24-03285-f003:**
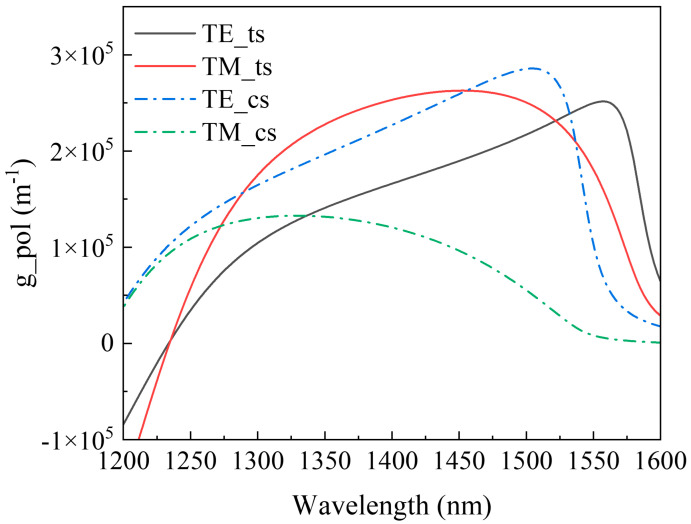
Strain quantum well material gain curves.

**Figure 4 sensors-24-03285-f004:**
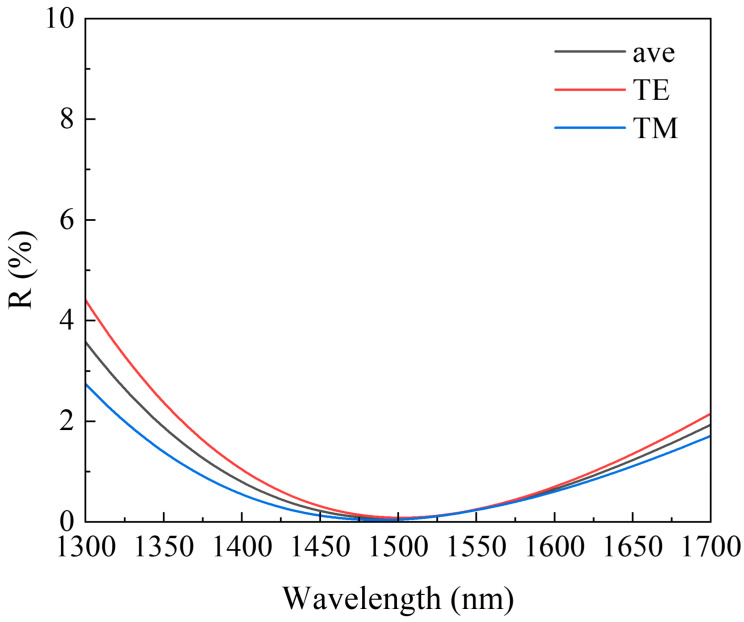
Simulation curves of energy reflectivity for different polarization states in a single-layer AR film.

**Figure 5 sensors-24-03285-f005:**
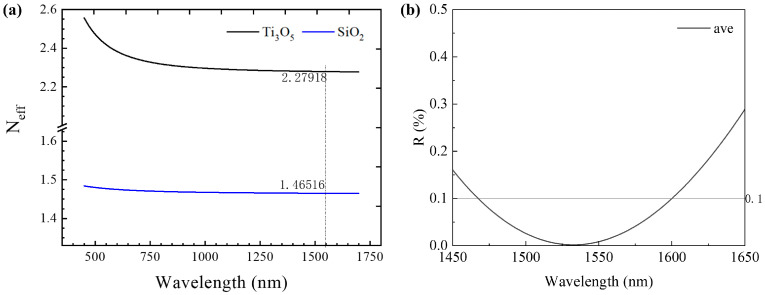
Ultra-broadband AR film: (**a**) material dispersion curve; (**b**) energy reflectivity simulation curve.

**Figure 6 sensors-24-03285-f006:**
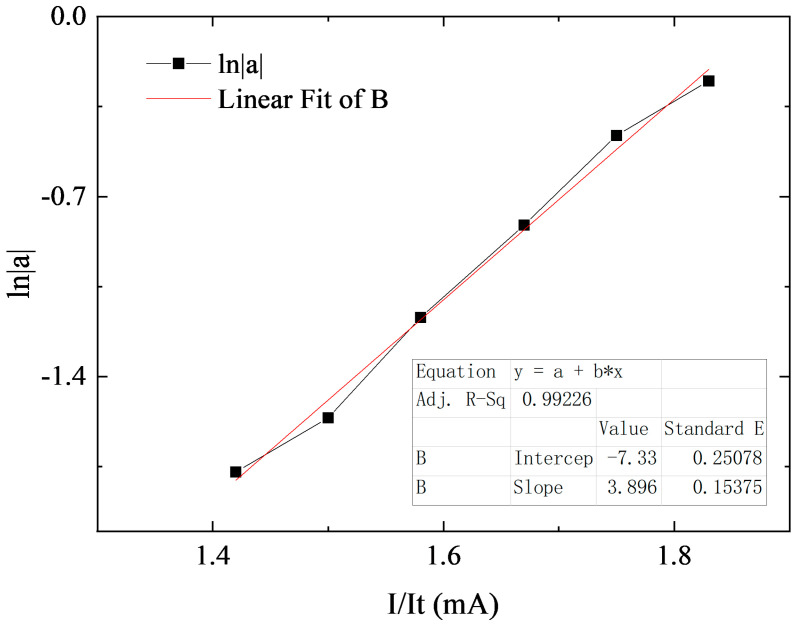
Relationship curve between I/I_t_ and ln|a|.

**Figure 7 sensors-24-03285-f007:**
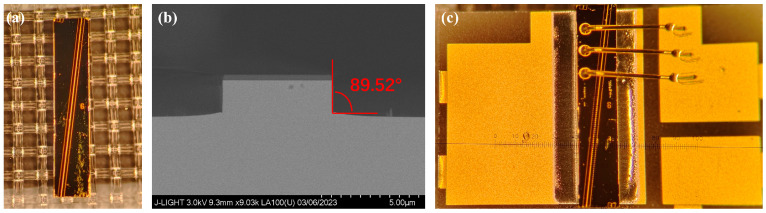
(**a**) Actual picture of SOA; (**b**) scanning electron microscope image of SOA cross-section; (**c**) COS package of SOA.

**Figure 8 sensors-24-03285-f008:**
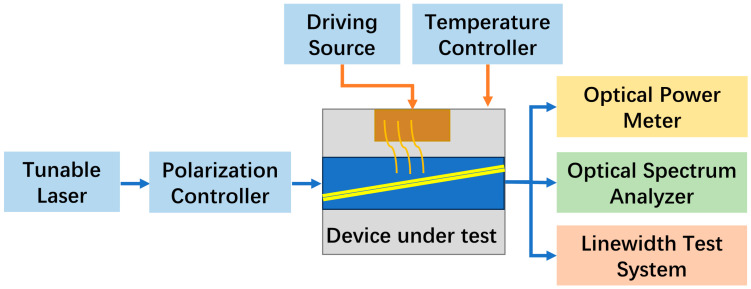
Schematic diagram of SOA test platform.

**Figure 9 sensors-24-03285-f009:**
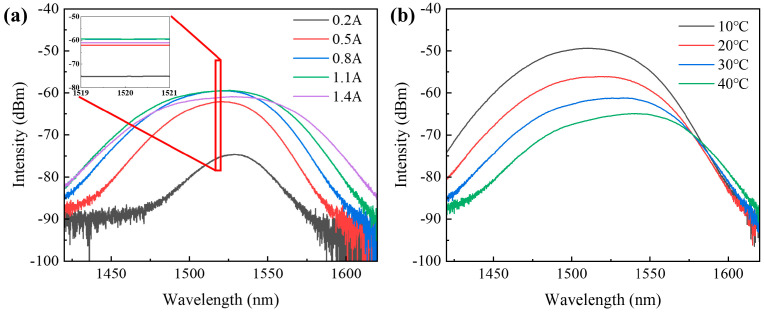
ASE curves of SOA under test: (**a**) at different currents; (**b**) at different temperatures.

**Figure 10 sensors-24-03285-f010:**
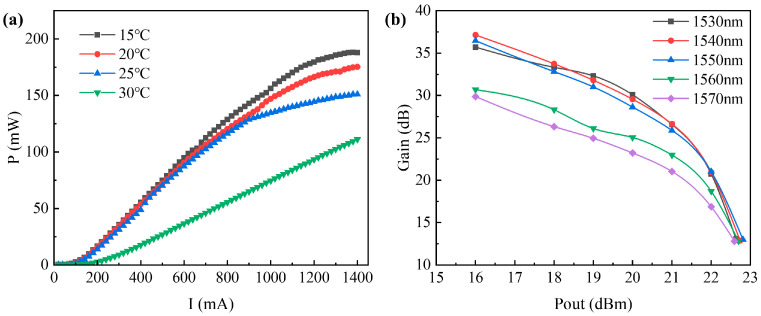
(**a**) Curves of current and output power of the SOA under test at temperature; (**b**) curves of output power and gain of the SOA under test.

**Figure 11 sensors-24-03285-f011:**
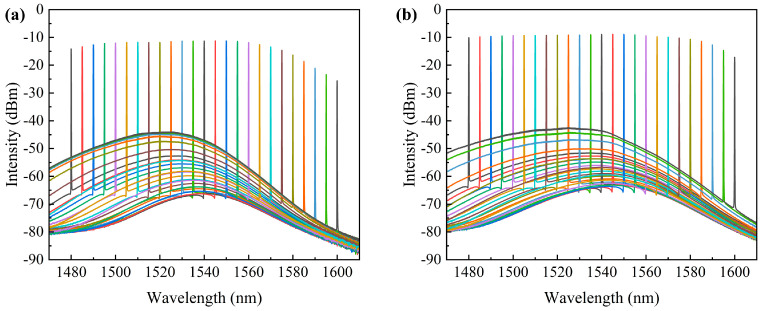
Amplified spectrum of SOA under test: (**a**) I = 0.5 A; (**b**) I = 1 A.

**Figure 12 sensors-24-03285-f012:**
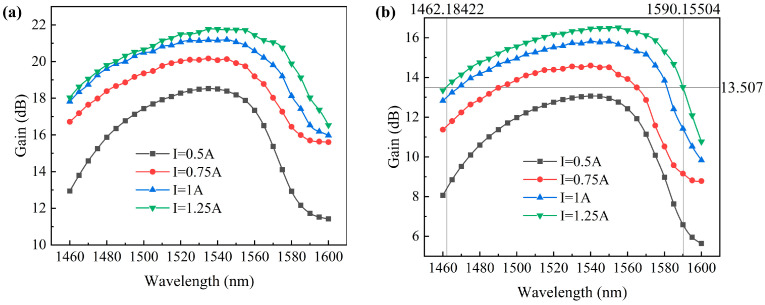
Gain curves of SOA to under test in the range of 1460–1600 nm with different driving currents: (**a**) laser power P_in_ = 0 dBm; (**b**) laser power P_in_ = 6 dBm.

**Figure 13 sensors-24-03285-f013:**
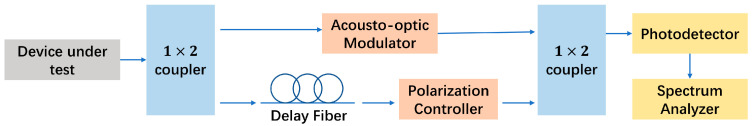
Schematic diagram of the delayed self-heterodyne linewidth measurement system.

**Figure 14 sensors-24-03285-f014:**
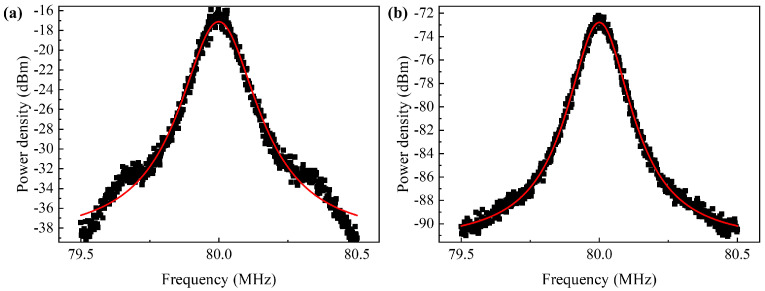
At 1550 nm band: (**a**) laser linewidth test result; (**b**) amplifier linewidth test result.

**Figure 15 sensors-24-03285-f015:**
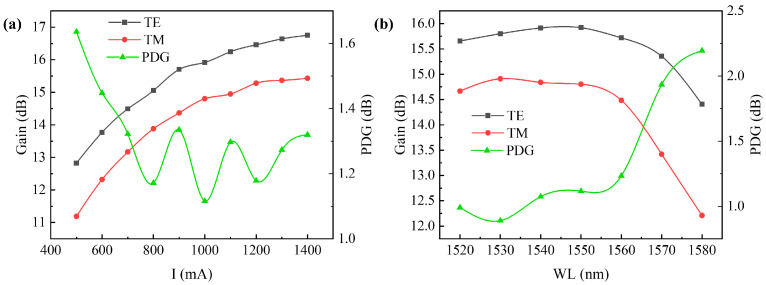
In the 1550 nm band: (**a**) relationship between PDG and current; (**b**) relationship between PDG and wavelength.

**Figure 16 sensors-24-03285-f016:**
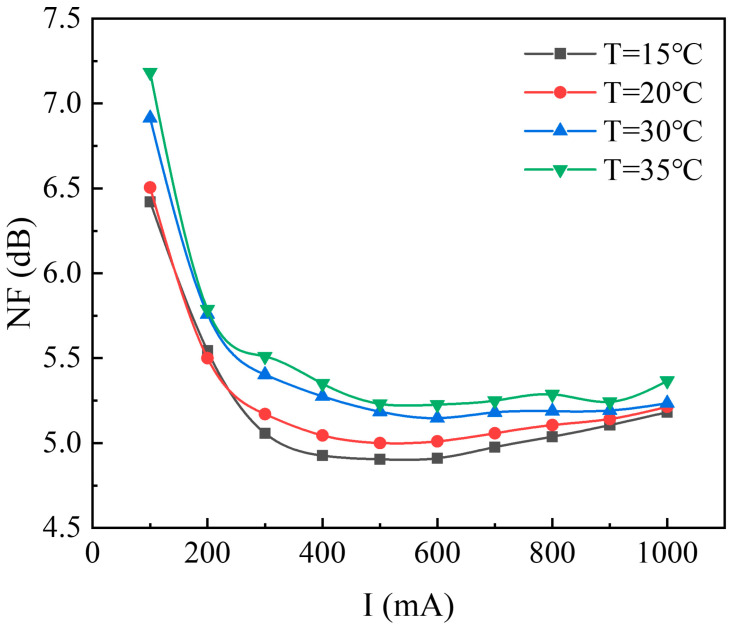
Noise figure curve.

**Table 1 sensors-24-03285-t001:** Detailed description of the SOA composition.

Layer	Material	Repeat	Thickness (nm)	Doping Concentration	CV Level (cm^−3^)
16	GaInAs	/	200	p-Zinc	>1.50 × 10^19^
15	GaInAsP	/	50	p-Zinc	>3.00 × 10^18^
14	InP	/	100	p-Zinc	>1.50 × 10^18^
13	InP	/	1500	p-Zinc	=1.00 × 10^18^
12	GaInAsP	/	20	p-Zinc	=1.00 × 10^18^
11	InP	/	50	p-Zinc	=7.00 × 10^17^
10	AlGaInAs	/	60	p-Zinc	=4.00 × 10^17^
9	AlGaInAs	/	60	Undoped	
8	AlGaInAs	/	10	Undoped	
7	AlGaInAs	5	6	Undoped	
6	AlGaInAs	5	10	Undoped	
5	AlGaInAs	/	60	Undoped	
4	AlGaInAs	/	60	n-Silicon	=1.00 × 10^18^
3	AlGaInAs	/	10	n-Silicon	=1.00 × 10^18^
2	InP	/	500	n-Silicon	=1.00 × 10^18^
1	InP	/	300	n-Silicon	=3.00 × 10^18^

**Table 2 sensors-24-03285-t002:** Comparison of SOA performance characteristics.

Reference	P (dBm)	Gain (dB)	Bandwidth (nm)	NF (dB)	PDG (dB)
Ref. [8]	15.8	14	90	6.5	0.8
Ref. [9]	19.6	15	120	4.5	/
Ref. [16]	/	21	/	/	~1.1
Ref. [17]	/	31.4	~50	/	0.8
Ref. [18]	21.5	~33	~60	/	0.4
Ref. [19]	21	21	~50	/	<2
Ref. [20]	/	20	100	/	<0.5
This paper	22.7	36.4	128	5.5	<1.4

## Data Availability

Data are contained within the article.

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
