# Peer review of "Semiconductor Optical Amplifiers with Wide Gain Bandwidth and Enhanced Polarization Insensitivity Based on Tensile-Strained Quantum Wells"

_sensors, 2024, doi:10.3390/s24113285_

Round 1

Reviewer 1 Report

Comments and Suggestions for Authors

Reviewer report:

The manuscript: Wide gain bandwidth and low polarization semiconductor optical amplifiers based on tensile-strained quantum wells is an interesting work in the field of quantum well optical devices. Manuscript describes a wide gain bandwidth, low polarization semiconductor optical amplifier based on tensile-strain quantum wells. The topic is relevant to the field as it presents improvements in the optical gain and broadening of the bandwidth. Authors use the combination of tilted waveguide design with cavity surface optical film design enabling effective minimization of the cavity surface reflectivity of the semiconductor optical amplifier. Methodology used is relevant to the problems to be solved. Conclusions are relevant to obtained results. References are appropriate.

I recommend to publish the manuscript after addressing following comments:

Page 4, line 154 - Put equation number at the end of line 153 to make it easy to find.

Page 4, line 159-160 - the ter. “e0 is the vacuum capacitance” is misleading.  Should be vacuum dielectric constant?

Page 5, line 171- Heavy hole band of the lepton? It is confusing. Why do not simply write "electron"?

Page 5, line 187 - It is not clearly written. Eq. 12 should also be refereed.

Figure 4 - Magnify scale on Y-axis. It will allow to better present the differences between the curves.

Figure 9 – Insert is barely visible. Expand it.

Figure 11  is not referred in the text. Also, it is unclear what vertical color likes depict.

Author Response

For research article

Response to Reviewer 1 Comments

1. Summary

Thank you very much for taking the time to review this manuscript. We have carefully revised the paper based on your suggestions and all revisions are marked in red. We will respond to your review comments one by one as following.

2. Questions for General Evaluation

Reviewer’s Evaluation

Response and Revisions

Does the introduction provide sufficient background and include all relevant references?

Yes

Thanks

Are all the cited references relevant to the research?

Yes

Thanks

Is the research design appropriate?

Yes

Thanks

Are the methods adequately described?

Can be improved

Have be improved

Are the results clearly presented?

Can be improved

Have be improved

Are the conclusions supported by the results?

Yes

Thanks

3. Point-by-point response to Comments and Suggestions for Authors

Comments 1: Page 4, line 154 - Put equation number at the end of line 153 to make it easy to find.

Response 1: Thank you for pointing this out. We agree with this comment. We resized the formulas to ensure that the formula numbers are on the same line. The specific modifications can be found on page 4, line 142.

Comments 2: Page 4, line 159-160 - the ter. “e0 is the vacuum capacitance” is misleading.  Should be vacuum dielectric constant?

Response 2: Agree. Thanks for pointing out the sloppy interpretation of constants, we have corrected it. The specific modifications can be found on page 4, line 147.

“e0 is the vacuum capacitance” is modified to “ε0 is the vacuum permittivity”.

Comments 3: Page 5, line 171- Heavy hole band of the lepton? It is confusing. Why do not simply write "electron"?

Response 3: We agree with this comment and we have corrected it. The specific modifications can be found on page 4, line 158-160, page 4, line 165-166, page 5, line 168-169, page 5, line 171-172

Lines 158 to 160: The matrix element for the transition from the electron with spin up <iS'| to the heavy hole band can be expressed as:

Line 165: the matrix element for the transition from the electron with another spin down <iS'| in the conduction band to the light hole band can be expressed as:

Line 168-169: the matrix element for the transition from the electron with spin up <iS'| to the heavy hole band can be expressed as:

Line 171-172: The matrix element for the transition from another spin-down electron <iS'| in the conduction band to the light hole band can be expressed as:

Comments 4: line 187 - It is not clearly written. Eq. 12 should also be refereed

Response 4: Thanks for your valuable comments. The equation has also be refereed. The specific modifications can be found on page 5, line 174.

Line 174: Substituting Equation (12), (14), (15), (16) back into Equation (8).

Comments 5: Figure 4 - Magnify scale on Y-axis. It will allow to better present the differences between the curves.

Response 5: Agree. We have adjusted the image Y-axis. The specific modifications can be found on page 7, line 241.

Comments 6: Figure 9 – Insert is barely visible. Expand it.

Response 6: Agree. We enlarged the nested picture. The specific modifications can be found on page 10, line 343.

Comments 7: Figure 11 is not referred in the text. Also, it is unclear what vertical color likes depict.

Response 7: Agree. We refer Figure 11 in the text and explain its contents. The specific modifications can be found on page 11, line 386-389.

Line 386-389: The output power of the seed source laser was set to 6 dBm. By varying the wave-length of the seed source laser from 1480 nm to 1600 nm in increments of 5 nm, the amplified spectra of the SOA were recorded. As shown in Figure 11, the overlaid schematic represents the amplified spectra across the entire test band.

4. Response to Comments on the Quality of English Language

5. Additional clarifications

Response 1: The English writing and presentation of the entire text has been further revised and improved.

Reviewer 2 Report

Comments and Suggestions for Authors

The paper developed a semiconductor optical amplifier (SOA) with wide gain bandwidth based on tensile-strain quantum wells specifically for the TM mode, in which the combination of tilted waveguide design with cavity surface optical film design is used to effectively minimize the cavity surface reflectivity of the SOA. However, the English writing and expression must further be improved for readability before being published. In addition, the following questions should also be clarified: (1) In the title of this paper, the expression of "low polarization" is improper because of the relative large polarization-dependent gain obtained from Table 1. (2) Please describe the structure of tensile-strained quantum wells for the homemade SOA. (3) The simulations on the SOA structure design in Section 2 should be associated with the fabricated SOA, specially for the key structural parameters or novel preparation techniques. (4) It is necessary to summary some design and preparation rules as well as the characterization of the SOA, instead of only describing the measured curves. (5) In some equations such as Eqs.(18) and (19), a number of parameters are not explained. However, Eq.(28) is not given in the paper, but is mentioned (Row 292). (6) Figs. 5(a) and 6 should be mentioned in the text. (7) How to obtain the curve given in Fig. 5(b)? (8) The expression of Eq. (14) is incorrect. (9) What is the on-chip modulation index method? (10) In Fig. 2, the values of n1 and n2 are not given and the simulated results are not identical with the SOA parameters, such as 7-degree tilted angle and 6um width.

Comments on the Quality of English Language

English writing and expression must further be improved for readability before being published.

Author Response

For research article

Response to Reviewer 2 Comments

1. Summary

Thank you very much for taking the time to review this manuscript. We have carefully revised the paper based on your suggestions and all revisions are marked in red. We will respond to your review comments one by one as following.

2. Questions for General Evaluation

Reviewer’s Evaluation

Response and Revisions

Does the introduction provide sufficient background and include all relevant references?

Yes

Thanks

Are all the cited references relevant to the research?

Yes

Thanks

Is the research design appropriate?

Yes

Thanks

Are the methods adequately described?

Can be improved

Have be improved

Are the results clearly presented?

Can be improved

Have be improved

Are the conclusions supported by the results?

Can be improved

Have be improved

3. Point-by-point response to Comments and Suggestions for Authors

Comments 1: In the title of this paper, the expression of "low polarization" is improper because of the relative large polarization-dependent gain obtained from Table 1.

Response 1: Thank you for pointing this out. We agree with this comment. We have designed our devices to have polarize below 1 dB in saturation, down to 0.89 dB. The title has been revised on page 1, lines 1-3.

Semiconductor optical amplifiers with wide gain bandwidth and enhanced polarization insensitivity based on tensile-strained quantum wells

Comments 2: Please describe the structure of tensile-strained quantum wells for the homemade SOA.

Response 2: Agree. We have added a further description of the quantum well and the thickness parameters of the layers. We apologize for the fact that the actual components of the quantum well cannot be provided directly due to internal requirements of the group. The specific modifications can be found on page 5-6, lines 201-203.

Table 1 presents the detailed epitaxial structure. The active region consists of 5 AlGaInAs quantum wells and 6 AlGaInAs barriers with a strain level of -0.21%. Positioned between the n-type AlGaInAs lower waveguide layer with a three-layer compositional gradient and the AlGaInAs upper waveguide layer with a two-layer compositional gradient, this active region structure has a distinctive feature: a lower optical field confinement factor, forming a weak waveguide structure. This feature facilitates two key actions: firstly, it broadens the optical field mode towards both the n-type lower waveguide layer and the p-type upper waveguide layer. Secondly, it deviates the central optical field mode away from the active region. Consequently, this configuration enhances the saturation output power.

Table 1. Detailed Description of the SOA Composition.

Layer

Material

Repeat

Thickness(nm)

Doping Concentration

CV level(cm-3)

16

GaInAs

/

200

p-Zinc

>1.50×1019

15

GaInAsP

/

50

p-Zinc

>3.00×1018

14

InP

/

100

p-Zinc

>1.50×1018

13

InP

/

1500

p-Zinc

=1.00×1018

12

GaInAsP

/

20

p-Zinc

=1.00×1018

11

InP

/

50

p-Zinc

=7.00×1017

10

AlGaInAs

/

60

p-Zinc

=4.00×1017

9

AlGaInAs

/

60

Undoped

8

AlGaInAs

/

10

Undoped

7

AlGaInAs

5

6

Undoped

6

AlGaInAs

5

10

Undoped

5

AlGaInAs

/

60

Undoped

4

AlGaInAs

/

60

n-Silicon

=1.00×1018

3

AlGaInAs

/

10

n-Silicon

=1.00×1018

2

InP

/

500

n-Silicon

=1.00×1018

1

InP

/

300

n-Silicon

=3.00×1018

Comments 3: The simulations on the SOA structure design in Section 2 should be associated with the fabricated SOA, specially for the key structural parameters or novel preparation techniques.

Response 3: Thank you for pointing this out. We have made modifications to the respective issues. The specific modifications can be found distributed on page 3 lines 110-119, page 5-6 lines 192-203, and page 5 lines 250-254.

page 3 lines 110-119:

Taking all these factors into consideration, we opted for the commonly used tilt angle of 7° for the waveguide, as reported in the literature. At this angle, under ideal conditions, the facet reflectance of waveguides with widths of 4 μm, 5 μm, and 6 μm can all be reduced to below 0.01%. In subsequent comparative experiments, we observed differences in the maximum output power among devices with waveguide widths of 4 μm, 5 μm, and 6 μm. Specifically, the device with a waveguide width of 4 μm had a maximum output power of approximately 70.6 mW, the device with a waveguide width of 5 μm had a maximum output power of approximately 139.3 mW, and the device with a waveguide width of 6 μm had a maximum output power of 188 mW. Consequently, we ultimately chose a waveguide structure with a width of 6 μm and a tilt angle of 7°.

page 5-6 lines 192-203:

Table 1 presents the detailed epitaxial structure, where the active region comprises 5 AlGaInAs quantum wells and 6 AlGaInAs barriers with a strain level of -0.21%. This active region is positioned between the n-type AlGaInAs lower waveguide layer with a three-layer compositional gradient and the AlGaInAs upper waveguide layer with a two-layer compositional gradient. The distinctive feature of this active region structure is its lower optical field confinement factor, forming a weak waveguide structure. This facilitates the broadening of the optical field mode towards the n-type lower waveguide layer and the p-type upper waveguide layer, deviating the central optical field mode away from the active region. Consequently, this configuration enhances the saturation output power.

Table 1. Detailed Description of the SOA Composition.

Layer

Material

Repeat

Thickness(nm)

Doping Concentration

CV level(cm-3)

16

GaInAs

/

200

p-Zinc

>1.50×1019

15

GaInAsP

/

50

p-Zinc

>3.00×1018

14

InP

/

100

p-Zinc

>1.50×1018

13

InP

/

1500

p-Zinc

=1.00×1018

12

GaInAsP

/

20

p-Zinc

=1.00×1018

11

InP

/

50

p-Zinc

=7.00×1017

10

AlGaInAs

/

60

p-Zinc

=4.00×1017

9

AlGaInAs

/

60

Undoped

8

AlGaInAs

/

10

Undoped

7

AlGaInAs

5

6

Undoped

6

AlGaInAs

5

10

Undoped

5

AlGaInAs

/

60

Undoped

4

AlGaInAs

/

60

n-Silicon

=1.00×1018

3

AlGaInAs

/

10

n-Silicon

=1.00×1018

2

InP

/

500

n-Silicon

=1.00×1018

1

InP

/

300

n-Silicon

=3.00×1018

page 5 lines 250-254:

The material system for the ultra-broadband AR film designed in this study employs Ti3O5 and SiO2, with the effective refractive indices of materials illustrated in Figure 5(a). The AR optical film structure was simulated and designed using tfcalc software, and its reflection simulation curve is depicted in Figure 5(b).

Comments 4: It is necessary to summary some design and preparation rules as well as the characterization of the SOA, instead of only describing the measured curves.

Response 4: Thanks for your valuable suggestion. We added some summaries of the design. The specific modifications can be found distributed on page 9 lines 307-310, page 10 lines 338-342, page 11 lines 377-383, page 11 lines 392-398, and page 12 lines 416-421.

page 9 lines 307-310:

The p-type electrode and n-type electrode of the SOA employ Ti/Pt/Au and Au/Ge/Ni/Au materials, respectively. As illustrated in Figure 7(c), the SOA is flip-chip bonded with its n-side facing downwards onto an aluminum nitride heat sink, establishing ohmic contact.

page 10 lines 338-342:

With the increase in current or temperature, the ASE spectrum underwent a redshift. As previously discussed during the quantum well design, the material gain center was biased towards shorter wavelengths for this reason. Following the redshift, the ASE spectrum center approached 1550 nm, consistent with expectations.

page 11 lines 377-383:

The low optical confinement factor quantum well structure we employed allows the optical field within the waveguide to expand to the upper and lower waveguide layers, successively accommodating more optical field energy. This design partially compensates for the low output power caused by the strained quantum well structure, resulting in a higher output power of the SOA compared to previously reported low-polarization SOAs. However, despite these improvements, the output power remains insufficient compared to strain-compensated quantum well SOAs with similar waveguide structures.

page 11 lines 392-398:

After amplification by the amplifier, the SMSR of the SOA remained above 50 dB throughout the entire tuning range. The spectra amplified by the SOA exhibited low background noise across the entire wavelength range of 1480-1580 nm, with smooth curves attributed to the combination of the tilted waveguide structure and the ultra-wideband AR optical thin film. This integration resulted in ultra-low reflectivity across the entire gain bandwidth, effectively suppressing cavity reflections and reducing spontaneous noise.

page 12 lines 416-421:

The device achieved a flat and wide gain spectrum, consistent with our expectations of extending the gain bandwidth by compensating for the device's gain in the short wavelength band when designing the strained quantum well in the previous section. The ultra-wideband AR optical film we designed completely covers the entire gain bandwidth, ensuring that the device's power is not lost due to cavity reflections within the gain bandwidth.

Comments 5: In some equations such as Eqs.(18) and (19), a number of parameters are not explained. However, Eq.(28) is not given in the paper, but is mentioned (Row 292).

Response 5: Thank you for your valuable suggestion. After careful consideration, we determined that the content related to Equation (18) is of little value and has been deleted. We have supplemented the text with the introduction of the parameters of Equation (19). Additionally, the reference to Equation (28) has been corrected. The specific modifications can be found on page 6 lines 217-219 and page 8 line 293.

page 6 lines 217-219:

where  is the phase thickness of the optical film, θ1 is the angle between the incident light and the normal to the incident plane, d1 is the optical thickness of the film, and nb is the effective refractive index of the optical film.

page 8 line 293:

By incorporating Equation (27)

Comments 6: Figs. 5(a) and 6 should be mentioned in the text.

Response 6: Agree. We have mentioned Figs. 5(a) and 6 in the text. The specific modifications can be found on page 7 lines 250-252 and page 8 lines 291-292.

page 7 lines 250-252:

The material system for the ultra-broadband AR film designed in this study employs Ti3O5 and SiO2, with the effective refractive indices of materials illustrated in Figure 5(a).

page 8 lines 291-292:

The results of straight-line fitting to the data are shown in Figure 6.

Comments 7: How to obtain the curve given in Fig. 5(b)?

Response 7: The curve given in Fig. 5(b) was simulated and designed using tfcalc software. The specific modifications can be found on page 7 lines 252-254.

The AR optical film structure was simulated and designed using tfcalc software, and its reflection simulation curve is depicted in Figure 5(b).

Comments 8: The expression of Eq. (14) is incorrect.

Response 8: Thanks for your valuable suggestion. Eq. (14) is referenced in “Physics of photonic devices” (Ref[14]). Considering the different definitions of the matrix elements of momentum operator Mb in the literature, Eq. 14 is deleted. The specific modifications can be found on page 4 line 164.

where m0 is the electron mass and Mb is the matrix elements of momentum operator

Comments 9: What is the on-chip modulation index method?

Response 9: The on-chip modulation index method is a technique used to estimate the reflectance of the AR optical film on its cavity surface using the gain model of a superluminescent diode (SLD). We have added specific details to the text. The specific modifications can be found on page 8 lines 262-280.

The on-chip modulation index method is a technique used to estimate the reflectance of the AR optical film on its cavity surface using the gain model of a super luminescent diode (SLD). The modulation index m of the SLD is expressed as:

where Pmax and Pmin represent the maximum and minimum values of the SLD output power density spectrum. Additionally, according to the gain model of SLD, m can be expressed as:

where a represents the round-trip gain factor, which can be expressed as:

where R1, R2 are the cavity surface reflectivity, L is the length of the laser. Gl is the single-pass gain, which can be expressed as:

where α is the loss coefficient, It is the threshold current of the laser, κ is the gain coefficient at It, From equations (24) and (25), we can obtain that the round-trip gain factor a can be expressed as:

where Rs is the cavity surface reflectivity in the case of uncoated surface, which is usually 0.32.

Comments 10: In Fig. 2, the values of n1 and n2 are not given and the simulated results are not identical with the SOA parameters, such as 7 degree tilted angle and 6 μm width.

Response 10: Thanks for your valuable suggestion. We added specific reasons for choosing 7° and 6μm ridge widths in the text. The specific modifications can be found on page 3 lines 99-119.

However, an excessively wide waveguide can also affect the single-mode performance of the SOA. To maintain single-mode operation, the etch depth of the waveguide is inversely related to the waveguide width. Deeper etch depths make it easier to encounter higher-order modes under the same ridge width conditions. Additionally, the etch depth also influences the optical field confinement factor of the active region. Shallow etch grooves can severely impact the power of the SOA. Taking into account the thicknesses of the active region, upper waveguide layer, and upper cladding layer, an appropriate etch depth for the epitaxial structure designed in this paper is around 1600 μm. This depth allows for a single-mode operation with a large ridge width of 6 μm, ensuring single-mode operation within acceptable error margins.

Taking all these factors into consideration, we opted for the commonly used tilt angle of 7° for the waveguide, as reported in the literature. At this angle, under ideal conditions, the facet reflectance of waveguides with widths of 4 μm, 5 μm, and 6 μm can all be reduced to below 0.01%. In subsequent comparative experiments, we observed differences in the maximum output power among devices with waveguide widths of 4 μm, 5 μm, and 6 μm. Specifically, the device with a waveguide width of 4 μm had a maximum output power of approximately 70.6 mW, the device with a waveguide width of 5 μm had a maximum output power of approximately 139.3 mW, and the device with a waveguide width of 6 μm had a maximum output power of 188 mW. Consequently, we ultimately chose a waveguide structure with a width of 6 μm and a tilt angle of 7°.

4. Response to Comments on the Quality of English Language

Point 1: English writing and expression must further be improved for readability before being published.

Response 1: Agree. The English writing and presentation of the entire paper has been further revised and improved.

5. Additional clarifications

None

Round 2

Reviewer 2 Report

Comments and Suggestions for Authors

The authors have completely revised the questions pointed out. I think the revision may be accepted if English language is improved again.